# How breakthroughs happen: Unearthing the boundary conditions of eco-friendly deliberate practice and eco-innovation performance

Yin-shi Jin[1], Asia Sohail[2], Shahid Iqbal[3], Tehreem Fatima[4], Arslan Ayub[5]*

1 College of Political and Law, Changchun Normal University, Changchun, China, 2 Faculty of Business and Management, Muscat University, Muscat, Oman, 3 Department of Project Management & Supply Chain Management, Bahria University, Islamabad, Pakistan, 4 Malik Firoz Khan Noon Business School, University of Sargodha, Sargodha, Pakistan, 5 National Business School, The University of Faisalabad, Faisalabad, Pakistan

* drarslanayub@gmail.com

**Data Availability Statement:** Data is attached within the manuscript and as a supplementary material.

## Abstract

Surprisingly innovation process based on deliberate practice has rarely been unearthed that might explore the boundary conditions of the eco-friendly deliberate practice and eco-innovation performance relationship. Anchored on the organizational support theory and the social cognitive, the current study seeks to investigate the impacts of perceived organizational support (POS) and developmental leadership (DL) on eco-innovation performance (EP) through the mediating role of eco-friendly deliberate practice (EDP). In addition, the study explores the boundary effects of employee resilience (ER) on the relationship between EDP and EP. The study collects time-lagged (i.e., "three-wave") and multisource (i.e., "self-rated and supervisor-rated") data from 383 respondents working in the service sector organizations in Pakistan. The authors processed data in SmartPLS (v 4.0) to assess the measurement model and the structural model. The study finds that POS and DL have significant positive relationships with EDP. Further, EDP partially mediates the links between POS, DL, and EP. Moreover, ER intervenes the association between EDP and EP such that at high levels of ER, the relationship is stronger and vice versa. Despite growing interest in deliberate practice, the boundary conditions of EDP in the work context are rarely investigated. This is the first study that explores the contextual and individual factors that can underpin the influence of EDP on EP.

## Introduction

In today's competitive and diverse work environment, the ability of employees to innovate is crucial for ensuring the continued development and sustainability of organizations [1, 2]. Not only is innovation performance required, but eco-innovation performance (EP)—the capacity to innovate in ways that contribute to environmental sustainability—has become a pressing

**Funding:** This article is one of the interim results of the Jilin Provincial Education Department's 2022 Fund Project "Research on the Construction System of Jilin Province's Long-Term Care Talent Team (JJKH20230862SK)" and the 2022 Changchun Normal University Humanities and Social Sciences Fund Project "Comparative Study on the Construction System of Long-Term Care Talent Teams in China and Japan (CSJJ2022010SK)". The funders had no role in the study design, data collection, analysis, decision to publish, or manuscript preparation.

**Competing interests:** The authors declare no conflict of interest.

necessity [3–6]. With growing academic focus and increasing demands for sustainable business practices, organizations now realize the importance of developing employees who can significantly contribute to eco-friendly innovation [7]. Transitioning business activities to eco-friendly processes is no longer optional; it is a critical mandate for organizations aiming to foster eco-innovation [8]. Eco-innovation is defined as "the production, assimilation, or exploitation of a product, process, service, or method that reduces environmental risks, pollution, and resource use throughout its lifecycle" [9] (p. 7). Given the importance of eco-innovation, organizations must focus on enhancing employees' EP, which positively impacts both individual performance and the organization's bottom line [10].

Despite the recognized importance of eco-innovation for organizational sustainability, there remains a lack of empirical studies exploring the boundary conditions that influence EP [11]. Recent studies have begun to investigate individual factors contributing to EP [12, 13], but contextual factors remain underexplored. This study addresses this gap by investigating the impact of perceived organizational support (POS) and developmental leadership (DL) on EP, with eco-friendly deliberate practice (EDP) as a mediator and employee resilience (ER) as a moderator. Specifically, the study proposes that (1) POS and DL influence EP, (2) EDP mediates the relationship between POS, DL, and EP, and (3) ER moderates the relationship between EDP and EP, making it stronger when ER is high and weaker when it is low.

Perceived organizational support (POS) reflects employees' perceptions of how much their organization values their contributions and cares for their well-being [14]. Grounded in organizational support theory [15], POS creates a reciprocal relationship, where employees, in turn, contribute to improving their performance. Similarly, developmental leadership (DL), which focuses on fostering employee growth through mentoring and guidance, is instrumental in enhancing EP [16]. These contextual factors, POS and DL, act as key drivers of EDP, a self-regulated and sustained behavior aimed at improving eco-innovation through deliberate effort [17]. Employees who perceive strong organizational and leadership support are more likely to engage in EDP, enhancing their eco-innovation performance [18].

In addition to these contextual factors, employee resilience (ER) plays a vital role in strengthening the relationship between EDP and EP. ER serves as a coping mechanism that prevents performance plateaus and allows individuals to adapt and excel [19, 20]. High levels of ER enable employees to push beyond the limitations of routine behaviors, maintaining the drive necessary to continue improving eco-innovation performance. As a result, ER enhances the positive effects of EDP on EP. This study fills a critical gap in the eco-innovation literature by examining how contextual (POS, DL) and individual (EDP, ER) factors interact to promote EP. By investigating these relationships, the study contributes to a better understanding of the mechanisms that drive eco-innovation in the workplace, providing actionable insights for organizations seeking to promote sustainable innovation.

## Theoretical framework

The proposed relationships in this study draw on multiple theoretical frameworks, each contributing unique insights into understanding how perceived organizational support (POS), developmental leadership (DL), and eco-friendly deliberate practice (EDP) influence eco-innovation performance (EP). By integrating social cognitive theory [21, 22], path-goal theory [23], and transformational leadership theory [24], a more comprehensive view emerges regarding how contextual and individual factors interact to drive eco-innovation outcomes.

Social cognitive theory [21, 22] posits that individuals act as agents who can influence their environment through deliberate and sustained actions. This theory supports the idea that employees' beliefs about their capacity to influence eco-friendly outcomes [25], bolstered by

POS and DL, drive their engagement in eco-friendly deliberate practice (EDP). POS and DL create an enabling environment, fostering the belief that employees can positively impact environmental performance through their efforts, in line with social cognitive theory's focus on self-efficacy and environmental control.

Path-goal theory [23] emphasizes the role of leadership in facilitating employee performance by providing guidance, support, and removing obstacles to goal achievement. Developmental leadership (DL) reflects the core premise of path-goal theory by highlighting the leader's role in nurturing employee development, which in turn encourages eco-friendly deliberate practice [26]. Leaders who focus on the development of their employees through coaching, feedback, and individualized consideration create a clear path toward the attainment of eco-innovation goals [26], aligning with both path-goal theory and social cognitive theory's emphasis on self-regulation and goal-directed behavior.

Transformational leadership theory [24] complements these perspectives by emphasizing the role of leaders in inspiring and motivating employees to exceed their self-imposed limits. Through transformational leadership behaviors—such as mentoring, offering vision, and instilling confidence—leaders can enhance employees' self-efficacy [27], encouraging them to engage in sustained deliberate practice that contributes to eco-innovation performance. The transformational approach connects directly to the social cognitive framework by elevating employees' beliefs in their ability to effect change through deliberate, goal-oriented actions.

In summary, the combination of social cognitive theory, path-goal theory, and transformational leadership theory creates a robust foundation for understanding the relationships in this study. While social cognitive theory provides the individual-level mechanism (self-efficacy and deliberate practice) for influencing eco-innovation, path-goal and transformational leadership theories explain the contextual support (POS and DL) that enables and enhances this process. Together, these frameworks provide a coherent explanation of how individual beliefs, supported by leadership and organizational practices, drive eco-innovation performance.

## Hypotheses

**Perceived organizational support and eco-friendly deliberate practice.** The OST posits that "employees develop global beliefs concerning the extent to which the organization values their contributions and cares about their wellbeing" [15] (p. 501). OST has garnered a burgeoning interest both in academia and practice due to its significant implications to view the employee-employer relationship from the employee perspective, *i.e.*, POS and its strong correlates with job satisfaction, affective organizational commitment, and other attitudinal consequences [28]. According to a recent meta-analytical review, POS is found to be strongly correlated with the antecedents *such as* working conditions, human resource practices, employee-organization context, leadership; and outcomes *including* well-being, employee performance, employee's orientation towards the organization and work [29]. Given the critical role of POS in influencing a wide range of individual's attitudes and behaviors, we postulate that POS has implications for employees' deliberate practice in organizations.

Deliberate practice refers to as "participation in intense, persistent, and highly focused efforts to improve current performance" [30] (p. 50), emerged as a special form of practice that aims to exaggerate performance targets [19]. Deliberate practice is a self-regulated, purposeful, and iterative process that embarks upon individual's dedication and commitment to extend their behavioral repertoire in the pursuit of excellence [17]. The conceptual roots of deliberate practice can be traced in the literature of sports [31] and instrumental music [32, 33]. Similarly, researchers have also investigated deliberate practice across a range of disciplines *such as* chess [31], artist performance [33], creative writing [34], insurance sales [35], software design [36],

and medicine and surgery [37]. Besides, the implications of deliberate practice on a variety of individuals' attitudes and behaviors are evident in extant literature [17, 19, 38]. With the growing popularity and eminent role in elevating individual performance, organizational scholars have recently realized the importance of deliberate practice in the work context [39].

When employees perceive that their organizations are concerned about environmental initiatives and value their contributions to promote environmentalism [40], they become motivated to deliberately engage in activities that enhance environmental performance. We draw on the insights from the social cognitive theory [21, 22], which posits that employees view themselves as organizational agents and they develop beliefs that their actions can significantly influence their environment. Therefore, they tend to invest their efforts and energy in pursuing goals that contribute to organizational environmental initiatives [41]. As EDP infuses purposeful and persistent employee behavior towards the attainment of these goals [8], they expect organizational support to fulfill these obligations [15]. Another argument that supports this assumption comes from the OST, which permeates a reciprocal obligation to move organizational endeavors forward by proactively partaking in activities, *i.e.*, EDP, that may result in accomplishing organizational goals and objectives. For instance, Levinson [42] sanctioned that organizational agents' actions are the reflection of organizational intent instead of their personal motives. This organizational personification "is abetted by the organization's legal, moral, and financial responsibility for the actions of its agents; by organizational policies, norms, and culture that provide continuity and prescribe role behaviors" (*e.g.*, facilitating employees' EDP) [28] (p. 698). Thus,

*H1a. POS positively influences EDP.*

**Developmental leadership and eco-friendly deliberate practice.** Zhang and Chen [16] defined DL as "supervisory behaviors aimed at developing subordinates' work-related knowledge and skills and facilitating their personal and vocational development". Despite its significant role in inspiring employees' performance outcomes, there is a scarcity of empirical research that found its positive correlates with individual's attitudes and behaviors [43]. A review of literature on DL submits several leadership behaviors that signposts individualized consideration *such as* "encourage followers to attend technical courses", "careful observation of staff", and "career counseling" [24]. Moreover, these behaviors take various forms *such as* offering developmental experiences, providing feedback, counseling, coaching, guiding, and mentoring [44]. Further, the magnitude and scope of these behaviors may vary depending on the developmental needs of the followers [16]. Guided by the central premise of DL to stimulate behavioral development in followers, we suggest that DL influences employees' EDP. Several prior studies have deliberated DL with commonly labeling it as DL [16]. For instance, House's [23] path goal theory of leadership offers supportive leadership style through guidance and coaching to the subordinates. Moreover, supervisory mentoring in DL is predominantly reflected in the mentoring literature [45]. Similarly, recognition of individual's needs for growth and achievement is rooted in the transformational leadership theory [46]. As a core of transformational behavior DL stimulates followers' self-efficacy and skills, and thus, has "*transformational effects*" [47]. This is in congruent with the social cognitive theory [21, 22] such as followers' self-efficacy enrichment through leader's developmental behaviors triggers the intent to engage in sustained behaviors, *i.e.*, EDP for accomplishing superior outcomes. In a related stream, Zhang and Chen [16] reported positive correlates between DL and organizational citizenship behaviors. Rafferty and Griffin [47] identify DL as a distinguished construct encouraging personal and vocational development of followers. These arguments support our theoretical deduction that DL nurtures employees' EDP through exercising behaviors such as

coaching, mentoring, teaching, and feedback, etc. Further, the link between coaching/mentoring and deliberate practice is also evident in prior studies [48]. Thus,

*H1b. DL positively influenced EDP.*

**Eco-friendly deliberate practice and eco-innovation performance.** We further expect that EDP elevates employees' EP. As Baker *et al.* [38] (p. 65) endorsed that EDP is "predicated on the concept that it is not simply training of any type, but the engagement in specific forms of practice (*e.g.*, eco-innovation), that is necessary for the attainment of expertise". Similarly, Bilal and Fatima [39]; Ericsson [49] argued that deliberate practice is a self-regulated, purposeful, and persistent individual's engagement in an activity, which elevates improvement in the current performance based on feedback and repetition of behavior. Extensive evidence suggests that continuous improvement in the practice (*e.g.*, EDP) leverages long-term performance (*e.g.*, EP) [17, 19, 50–52]. In addition, the relationship between EDP and EP has been verified in a recent study by Miao *et al.* [8], who found that EDP influences EP through the mediating role of creative self-efficacy. Thus,

*H2. EDP positively influences EP.*

**Mediating role of eco-friendly deliberate practice.** Taken together, these arguments suggest the mediating role of EDP between the underlying linkages. Guided by the social cognitive theory [21, 22], employees' perception of favorable support from organizations and leaders, in terms of POS and DL, stimulates their behavioral tendency to manifest EDP, ultimately fostering EP. This is because, EDP engenders individuals to elicit task performance through persistence and sustained engagement in eco-friendly activities, which in turn, elevates their confidence in mastering the ability to perform the task, thereby, facilitating excellence in EP. Thus,

*H3a. EDP mediates the relationship between POS and EP.*

*H3b. EDP mediates the relationship between DL and EP.*

**Moderating role of employee resilience.** Although deliberate practice leverages enhanced performance through escalated beliefs (self-efficacy) and confidence in oneself to deliver superior outcomes. In juxtaposition, scholars also argue that increased levels of self-efficacy may sometimes lead individuals to get caught in the "arrested development" because of *routinization* and *automization* of the activity [17, 19]. Nevertheless, individuals' need to do modifications in their cognitive mechanisms in order to promote performance levels [19]. However, individuals with high self-efficacy fostered due to mastery experiences endure to contribute efforts in the prevailing tasks, which limits their ability to invest efforts in the succeeding levels [17]. Hence, repeating the same activity at the point of *arrested development* and *routinization* offers limited explanation to enhance performance targets. Therefore, employees exercising EDP may also experience depreciated levels of EP. As a coping mechanism, we suggest that ER is a crucial factor that stimulates the influence of EDP on EP. ER is defined as "behavioural capability to leverage work resources in order to ensure continual adaptation, well-being, and growth at work, supported by the organization" [53] (p. 224). Researchers argue that ER is one of the four key components of psychological capital, which refers to "those psychological assets or resources that allow individuals to thrive and prosper in the workplace by being positive" [54, 55]. ER reflects a growth mindset enabling employees to *bounce back* in critical situations

by preparing them to deal with personal setbacks or adversities and managing the associated risks [56]. In the similar milieu, Ahmad *et al.* [57] permitted that ER enhances individual's ability to effectively respond to challenging situations. Correspondingly, [58] have found that ER allows individuals to stay motivated and work hard when faced with difficulties. Likewise, Shin *et al.* [59] reported that ER prompts individual's ability to pull back from disruptions in functioning. Hence, ER might serve as a personal resource that enhances employees' self-regulatory cognitions and buffers negativities associated with high levels of self-efficacy, stimulating the link between EDP and EP. Thus,

*H4. ER moderates the relationship between EDP and EP such that the association is stronger (weaker) at higher (lower) levels of ER.*

## Conceptual framework

The conceptual framework of the study is illustrated in Fig 1 below:

## Method

### Participants and procedure

The study aims to investigate the impact of individual, *i.e.*, EDP, and contextual factors, *i.e.*, POS and DL as the key contributors to employees EP. Based on the purpose of the study, the authors employed a positivist philosophical stance using a quantitative approach to empirically assess the hypothesized relationships. Accordingly, the authors collected data from the target respondents using a face-to-face mode through purposive sampling technique [60]. Purposive sampling technique is useful when the study aims to achieve its purpose [60]. Besides, purposive sampling technique is appropriate because of its ability to yield the arbitrary response.

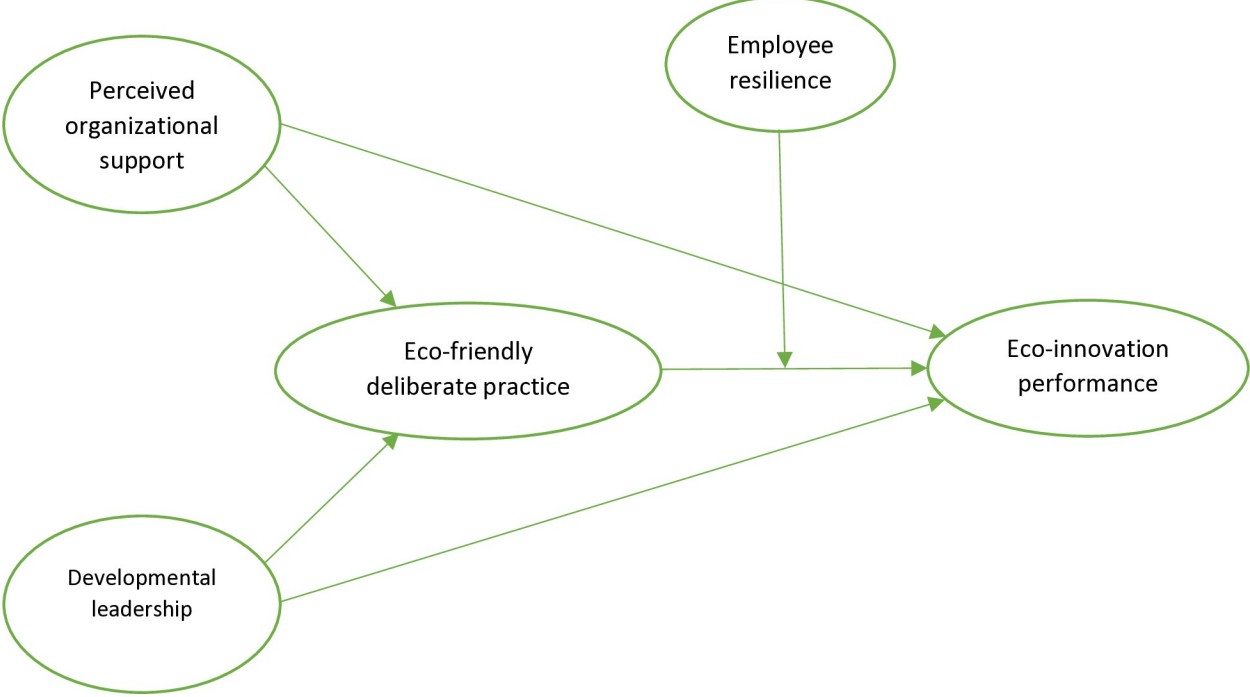

**Fig 1. Conceptual model.**

Through purposive sampling technique, the authors collected data from service employees in Pakistan in multiple waves. Cole and Maxwell [61] and Maxwell and Cole [62] suggested to employ a time-lagged research design when the model involves a causal mechanism and failure to do so may lead to possible biases in evaluating the parameter estimates. In addition, the authors conducted a multisource survey to minimize potential common method biasness (CMB) issues, *i.e.*, the survey included the self-rated and supervisor-rated responses.

Given the purpose of the study, the selected population is expected to be a true representative of the study because service industry faces an escalated competition and to transform sustainable business practices has been a key mandate of these companies [63, 64]. Several prior studies have surveyed the service sector employees to investigate innovation performance both at individual and organizational levels [65, 66].

To ensure consistency and accuracy in the data collection process, the research team received standardized training on how to distribute and explain the questionnaires to respondents. A clear protocol was established to ensure that respondents understood the study's purpose and were aware of the confidentiality of their responses. Instructions were also provided on how to handle any queries from respondents regarding the content of the questionnaire, ensuring that responses remained unbiased and accurate. To enhance the reliability of the data collection process, field supervisors were assigned to monitor the distribution of the questionnaires and the collection of responses. These supervisors conducted random checks to ensure adherence to the standardized process and that no deviations occurred during questionnaire completion. Additionally, all respondents were provided with a cover letter explaining the purpose of the study and instructions for creating personal identification keys to match responses across the three waves of data collection. The participants were provided questionnaires and cover letter. The cover letter contained important information such as the purpose of the study, ensuring confidentiality of their responses, and informed the procedure to generate keys so that the responses gathered across waves can be matched and consolidated. In the first wave, the authors distributed 500 questionnaires to the target respondents to collect responses for POS, DL, and ER. Moreover, the participants were also requested to provide information about their demographics and generate keys by using the first letters of their initials followed by their birth months. The respondents returned 465 questionnaires and after screening 434 questionnaires were redistributed in the second wave to gather responses for EDP. Of which, 411 respondents returned the questionnaires. In the final wave, we contact their supervisors to rate about their EP.

Finally, the authors consolidated all the responses collected in each wave and processed 383 completely filled questionnaires. The survey included both male (54%) and female (46%) participants with a mean age of 36.49 (SD: 5.24). Regarding their education, 27%, 41%, and 33% of the participants were "undergraduate", "graduate", and "postgraduate". They were also requested to provide information about their length of service. The survey revealed the following statistics: the participants' tenure was 18% (less than 1 year), 21% (between 1 to 3 years), 24% (between 3–5 years), 17% (between 5–8 years), and 20% (more than 8 years). Moreover, 53% and 47% of the participants worked in private and public sector organizations.

The study is in accordance with the ethical standards of the QEC of The University of Faisalabad, Faisalabad, Pakistan. The authors' received written consent of the participants for their voluntary participation in the survey. Participants' detailed information was not accessible, therefore, the authors selected purposive sampling approach for data collection started from August 2022 to January 2023. As discussed above, the participants were requested to generate keys so that there responses should be matched and consolidated.

## Measures

The authors adapted established measurement scales to determine the proposed model (see S1 Appendix). The survey items to assess POS were adapted from Eisenberger *et al.* [15], and contained 8 items. The sample items include "My organization strongly considers my goals and values". The survey items for DL were adapted from House's [44] developmental orientation, and contained 3 items. The sample items include "Encourages staff to improve their job-related skills". The survey items to measure EDP were adapted from Sonnentag and Irion [67], and contained 15 items. The sample items include "In order to improve my eco-innovation skills, I deliberately take some time to re-think my working technique". The survey items to measure EP were adapted from [68], and contained 4 items. The sample items include "Creates better eco-friendly processes and routines". The survey items to assess ER were adapted from Al-Omar *et al.* [69], and contained 6 items. The sample items include "I tend to bounce back quickly after hard times".

## Results

### Measurement model

The current study assessed the proposed model using the Structural Equation Model (SEM) in SmartPLS (v 4.0). The authors obtained the structural paths in two stages. In the first stage, the measurement model was assessed and the structural model was assessed in the second stage. This is in congruent with the recommendations of Hair *et al.* [70] to validate the measurement model to qualify data for further analysis. The measurement model assessed the reliability and validity of the study. For reliability analysis, Cronbach's alpha and composite reliability (CR) metrics were tested. According to Hair *et al.* [70] and Cronbach [71], the values of CR and Cronbach's alpha should range between 0.70 and o.95. The analysis reported in Table 1 illustrate that all the values are greater than the acceptable threshold, ensuring reliability of the data. Further, the convergent validity was assessed using the outer loadings and average variance extracted (AVE) criteria [70]. The findings show that all the values are greater than the acceptable threshold of 0.50 [70, 72], establishing convergent validity in the study. Besides, the followings items: POS7, EP3, EDP8, EDP10, EDP12 were eliminated due to poor loadings [70].

After validating the reliability and convergent validity, the authors measured the discriminant validity using the Fornell-Larcker and heterotrait-monotrait (HTMT) metrics (Table 2) [70, 73]. Fornell-Larcker criterion indicates the square root of the AVE values and specifies that the values of Fornell-Larcker for each construct are greater for the inter-construct correlation than for the outer-construct correlation [74]. In addition to the Fornell-Larcker metric, the authors also examined the HTMT ratio by running the bootstrapping procedure using the bias-corrected and accelerated (BCa) technique with 5,000 resamples. The analysis was assessed as a 90% significance level using one-tailed analysis to warrant an error probability of 95% at two-tailed and indicate that all the values are lower than the maximum threshold of $HTMT_{.85}$, establishing discriminant validity in the study. This is consistent with the recommendations of Henseler *et al.* [75].

### Structural model

Further, the authors assessed the structural model to analyze path coefficients ($\beta$)", "coefficient of determination ($R^2$), "predictive relevance" ($q^2$), and "effect size" ($f^2$). Table 3 shows the results of the structural model assessment. The findings indicate that POS ($\beta = 0.551$; $t = 11.983$; $p = 0.000$; $f^2 = 0.542$) and DL ($\beta = 0.298$; $t = 5.709$; $p = 0.000$; $f^2 = 0.158$) are

**Table 1. Validity and reliability for constructs.**

|  | Loadings | AVE | CR | Cronbach's alpha |
|---|---|---|---|---|
| Perceived organizational support |  | 0.538 | 0.890 | 0.857 |
| POS1 | 0.646 |  |  |  |
| POS2 | 0.636 |  |  |  |
| POS3 | 0.764 |  |  |  |
| POS4 | 0.777 |  |  |  |
| POS5 | 0.745 |  |  |  |
| POS6 | 0.816 |  |  |  |
| POS8 | 0.734 |  |  |  |
| Developmental leadership |  | 0.640 | 0.842 | 0.719 |
| DL1 | 0.788 |  |  |  |
| DL2 | 0.798 |  |  |  |
| DL3 | 0.814 |  |  |  |
| Eco-friendly deliberate practice |  | 0.505 | 0.917 | 0.900 |
| EDP1 | 0.725 |  |  |  |
| EDP11 | 0.746 |  |  |  |
| EDP13 | 0.606 |  |  |  |
| EDP14 | 0.626 |  |  |  |
| EDP2 | 0.624 |  |  |  |
| EDP3 | 0.741 |  |  |  |
| EDP4 | 0.632 |  |  |  |
| EDP5 | 0.840 |  |  |  |
| EDP6 | 0.833 |  |  |  |
| EDP7 | 0.713 |  |  |  |
| EDP9 | 0.685 |  |  |  |
| Employee resilience |  | 0.514 | 0.862 | 0.808 |
| ER1 | 0.520 |  |  |  |
| ER2 | 0.691 |  |  |  |
| ER3 | 0.675 |  |  |  |
| ER4 | 0.802 |  |  |  |
| ER5 | 0.768 |  |  |  |
| ER6 | 0.805 |  |  |  |
| Eco-innovation performance |  | 0.571 | 0.798 | 0.732 |
| EP1 | 0.670 |  |  |  |
| EP2 | 0.811 |  |  |  |
| EP4 | 0.778 |  |  |  |

*Notes*. POS: perceived organizational support; DL: developmental leadership: EDP: eco-friendly deliberate practice; EP: eco-innovation performance; ER: employee resilience; CR: composite reliability; AVE: average variance extracted

significantly and positively related to EDP, supporting *H1a* and *H1b*. Moreover, the effect size reveals the large and medium effects for the relationship between POS and EDP, and DL and EDP respectively. In addition, the findings indicate the significant positive relationship between EDP and EP ($\beta = 0.226$; $t = 2.350$; $p = 0.019$; $f^2 = 0.126$), with a medium effect size (supporting *H2*). The graphical illustration of the SEM is shown in Fig 2.

Moreover, the study proposed the mediating role of EDP between POS and DL and EP. For yielding the point estimates of the mediation analysis, the authors run the BCa bootstrapping technique involving a resample of 5,000 at a 95% significance level (two-tailed) [76]. Results of

**Table 2. Discriminant validity.**

| Fornell-Larcker | | | | | | Heterotrait-monotrait (HTMT) Ratio | | | | | |
|---|---|---|---|---|---|---|---|---|---|---|---|
| | DL | EDP | EP | ER | POS | | DL | EDP | EP | ER | POS |
| DL | 0.800 | | | | | DL | | | | | |
| EDP | 0.447 | 0.711 | | | | EDP | 0.553 | | | | |
| EP | 0.352 | 0.360 | 0.755 | | | EP | 0.520 | 0.446 | | | |
| ER | 0.244 | 0.532 | 0.291 | 0.717 | | ER | 0.328 | 0.624 | 0.362 | | |
| POS | 0.272 | 0.632 | 0.326 | 0.620 | 0.734 | POS | 0.343 | 0.697 | 0.407 | 0.624 | |

*Notes*: POS: perceived organizational support; DL: developmental leadership: EDP: eco-friendly deliberate practice; EP: eco-innovation performance; ER: employee resilience; CR: composite reliability

this analysis are shown in Table 4. The analysis indicates that both the total effects for POS → EP (CIs 0.051, 0.584) and DL → EP (CIs 0.183, 0.403), and the specific indirect effects for POS → EDP → EP (CIs 0.016, 0.240) and DL → EDP → EP (CIs 0.008, 0.129) are significant as the confidence intervals didn't straddle 0. In addition, the authors also examined the variance accounted for (VAF) and the values of 46.44% and 23.10% indicate the partial mediating role of EDP between POS and EP and DL and EP respectively. This analysis renders support to the hypotheses *H3a* and *H3b*.

Furthermore, the study examined the moderating role of ER on the relationship between EDP and EP using a two-stage moderation approach [70]. This is in consistent with the suggestions of Henseler and Fassott [77] to employ the two-stage moderation approach because of its superior power in predicting the association. Table 3 shows the result of moderation the moderation analysis. The findings illustrate that the interaction effect between ER and EDP on EP is significant (CIs 0.048, 0.152). Further, the effect size shows the medium effect of this relationship. Moreover, the authors assessed the graphical representation of the moderation effect by using the simple slope analysis (Fig 3) [78]. The average of the moderation effect is shown in the middle line. The other two lines represent the association between EDP and EP, *i.e.*, mean value plus one SD for higher levels of ER and mean value minus one SD for lower levels of ER, supporting *H4*.

In addition, following the recommendations of Tenenhaus *et al.'s* [79], we measured the goodness-of-fit (GOF) index. GOF refers to "the geometric mean of the average communality and average $R^2$" [79]. Results of this analysis are reported in Table 5 which indicate a good model fitness as the value of 0.436 is greater than the cutoff value of 0.36 for the large effect size of $R^2$ [80]. Finally, the predictive relevance of the proposed model was assessed using the Stone-Geisser's $Q^2$. The values above 0 ensured the predictive relevance of the study.

**Table 3. Effects on endogenous variables.**

| Hypotheses | β | CI (5%, 95%) | SE | *t*-value | *p*-value | Decision | $f^2$ | $R^2$ | $Q^2$ |
|---|---|---|---|---|---|---|---|---|---|
| *H1a* POS → EDP | 0.551*** | (0.464, 0.646) | 0.046 | 11.983 | 0.000 | Supported | 0.542 | 0.481 | 0.245 |
| *H1b* DL → EDP | 0.298*** | (0.192, 0.401) | 0.052 | 5.709 | 0.000 | Supported | 0.158 | | |
| *H2* EDP → EP | 0.226*** | (0.031, 0.400) | 0.096 | 2.350 | 0.019 | Supported | 0.126 | 0.205 | 0.375 |
| *H4* EDP x ER → EP | 0.077*** | (0.048, 0.153) | 0.048 | 2.581 | 0.007 | Supported | 0.116 | | |

*Notes*. POS: perceived organizational support; DL: developmental leadership: EDP: eco-friendly deliberate practice; EP: eco-innovation performance; ER: employee resilience

***significance p < 0.05 (1.96)

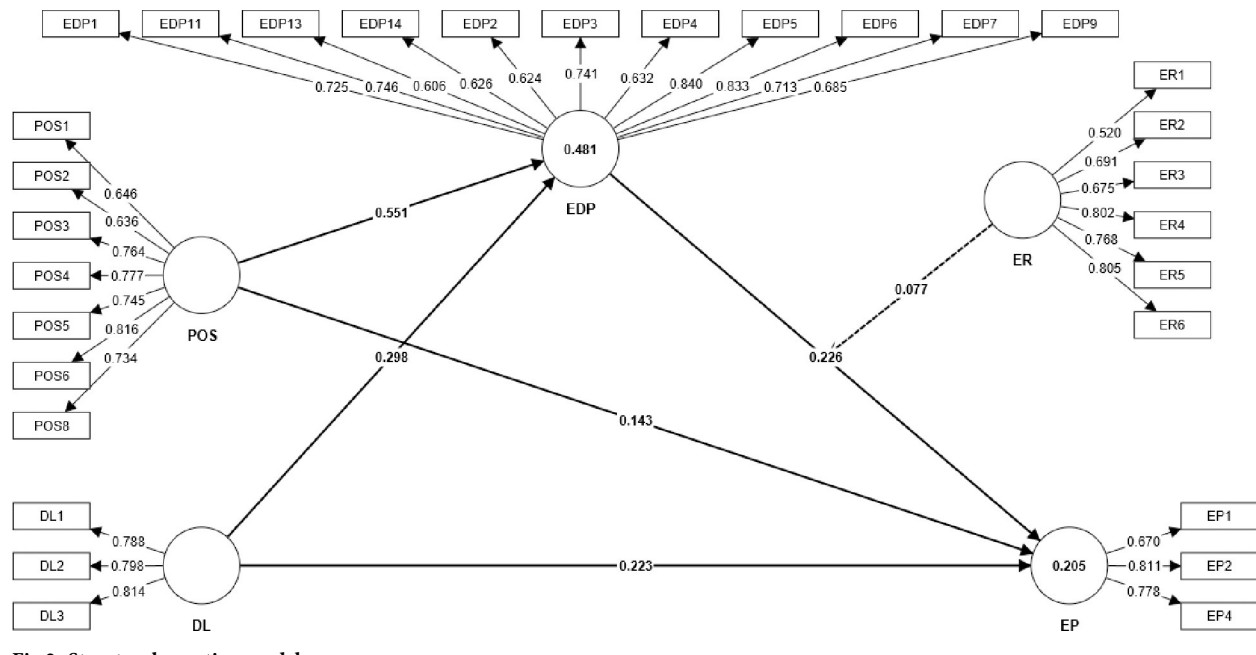

**Fig 2. Structural equation model.**

## Discussion and conclusion

Given the crucial role of employees in prompting eco-innovation in the service industry, EDP is an important yet underexplored construct that is capable of producing elevating EP. Thus, the main contribution of this study is to assess a hitherto unexplored mechanism that investigates the antecedents and outcomes of EDP in the service industry in Pakistan. Drawing on the organizational support theory and the social cognitive theory, the authors proposed that POS and DL function as the contextual precursors that are important in leveraging EP through the mediating role of EDP. Moreover, the authors also suggest that ER intervenes the association between EDP and EP such that it buffers the periled effects of ESE and stimulates individual's beliefs to effectively transform EDP into enhanced EP. The authors examined the hypothesized model on data from the service sector employees using a "time-lagged", "multi-source" research design. The findings support all the proposed hypotheses such that:

- POS and DL have significant and positive relationships with EDP;

- EDP significantly and positively elicits EP;

- The relationships between POS and EP and DL and EP are mediated by EDP; and

**Table 4. Summary of mediating effect tests.**

|  | Path | t-value | BCCI |  | Path | t-value | 95% BCCI | Decision | VAF |
|---|---|---|---|---|---|---|---|---|---|
| Total effect |  |  |  | Indirect effect |  |  |  |  |  |
| POS → EP | 0.267*** | 2.444 | (0.051, 0.584) | *H3a* POS → EDP → EP | 0.124*** | 2.239 | (0.016, 0.240) | Supported | 46.44% |
| DL → EP | 0.290*** | 5.215 | (0.183, 0.403) | *H3b* DL → EDP → EP | 0.067*** | 2.155 | (0.008, 0.129) | Supported | 23.10% |

*Notes*. POS: perceived organizational support; DL: developmental leadership: EDP: eco-friendly deliberate practice; EP: eco-innovation performance

***significance p < 0.05 (1.96)

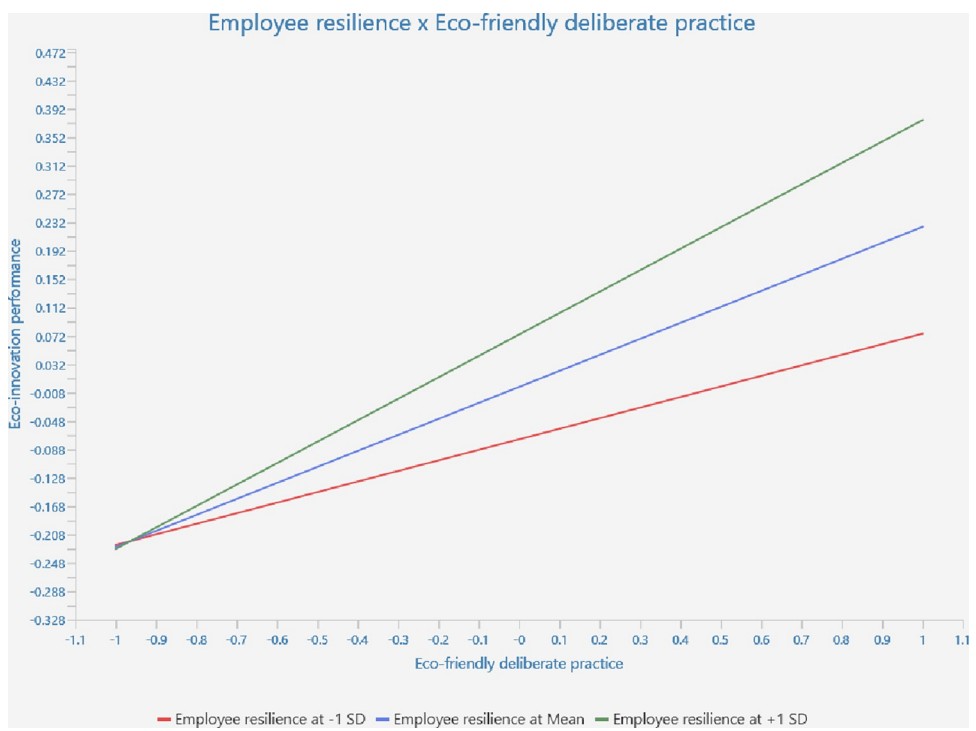

**Fig 3. Interaction effect of employee resilience and eco-friendly deliberate practice on eco-innovation performance.**

- ER moderates the association between EDP and EP such that at high levels of ER the relationship is more pronounced and vice versa.

The positive relationship between POS and EDP aligns with previous studies that emphasize the critical role of organizational support in fostering employees' performance and engagement in sustainable practices [14, 40]. Similarly, the significant link between DL and EDP is consistent with prior research on developmental leadership, which has been shown to enhance employee performance through coaching, mentoring, and skill development [16, 47]. These results reinforce the importance of leadership that promotes employee growth as a driver of eco-friendly behaviors. The mediating role of EDP between POS/DL and EP also supports the findings of Miao et al. [8], who found that deliberate practice fosters innovation performance through creative self-efficacy. However, this study extends their findings by focusing

**Table 5. Goodness-of-Fit Index (GFI).**

| Constructs | AVE | $R^2$ |
|---|---|---|
| DL | 0.640 | |
| EDP | 0.505 | 0.481 |
| EP | 0.571 | 0.205 |
| ER | 0.514 | |
| POS | 0.538 | |
| Average scores | 0.554 | 0.343 |
| $(GFI = \sqrt{\overline{AVE} \times \overline{R^2}})$ | 0.436 | |

*Notes*. POS: perceived organizational support; DL: developmental leadership: EDP: eco-friendly deliberate practice; EP: eco-innovation performance; ER: employee resilience; AVE: average variance extracted

specifically on eco-innovation and highlighting how deliberate eco-friendly practices contribute to organizational sustainability efforts. Moreover, the moderating role of ER in the EDP-EP relationship is consistent with research on resilience as a personal resource that enhances performance in challenging environments [53, 57]. The finding that higher levels of resilience strengthen the EDP-EP link suggests that resilient employees are better equipped to maintain their eco-innovation efforts despite potential setbacks, thus amplifying the effects of deliberate practice.

## Theoretical implications

The findings of the study offer several meaningful theoretical implications. First, the study postulates that for organizations to leverage sustainable competitive advantage, they require from their employees to proactively involve in the eco-friendly business activities. To promote eco-friendly initiatives, employees need to purposefully and persistently engage in behaviors that stimulate EP. Therefore, the authors proposed the eminent role of EDP as a significant factor in translating organizational environmental initiatives into sustainable business practices. Despite the critical role of deliberate practice in exaggerating individual's attitudes and behaviors, only a scarce sample of studies has investigated the implications of deliberate practice in the work context. For instance, prior research has investigated the role of deliberate practice in informal learning and entrepreneurial success [81], entrepreneurial learning and self-efficacy [82], and entrepreneurial expertise [83], among others. Further, this is one of the few efforts, with the valuable exception of Miao *et al.* [8], that cast deliberate practice as an effective internal resource to elevate individual's performance, particularly linking EDP with EP. By investigating EDP as an important resource, our study contributes to the extant literature of deliberate practice in the general context by entrenching its implications in the work context.

Second, the authors draw on the OST to predict that POS and DL facilitates EDP. To the best of the authors' knowledge, this is the first study that has explored the contextual antecedents of deliberate practice in the work context. For instance, Miao *et al.'s* [8] study assessed the role of EDP utilizing the individual level constructs and tested its relationship with EP through the mediating role of creative self-efficacy. By linking POS and DL as the external resource and EDP as an internal resource in organization, we demonstrated that EDP stimulated by POS and DL accomplishes organizational goals and objectives through prompting EP. Our findings suggest that POS provokes reciprocal behaviors among employees and they feel obligated to expand their behavioral repertoire by executing EDP, and hence it offers vital implications for the POS literature. Moreover, our study contributes to the limited literature on DL by linking it with EDP. Our findings endorse that DL infuses individualized consideration through mentoring and coaching that enact transformational effects [47]. Further, we also suggest that other leadership approaches such as transformational leadership theory [46], path goal theory [23], and individualized consideration [47] might also have implications for EDP. Both path goal theory and transformational theory have implications for the individualized consideration, which is imperative to foster EDP. We, therefore, invite future studies to employ these theoretical lens to extend the boundary conditions of the EDP–EP relationship. The combination of these findings corroborate the mediating role of EDP between POS and EP and DL and EP. We draw on the social cognitive theory [21, 22] and predicted the mediating role of EDP between the underlying links. The theory posits that employees accumulate mastery experiences through iterative, intentional, and self-regulated manner, which provokes self-efficacy, and thus, enhances eco-friendly performance outcomes, *i.e.*, EP.

At the core of SCT is the concept of *self-efficacy*, which refers to individuals' beliefs in their capabilities to execute behaviors necessary to achieve desired outcomes. In this study,

employees' perceptions of POS and DL contribute to the development of self-efficacy by fostering a supportive and empowering environment. This support helps employees accumulate *mastery experiences*, which are critical for building confidence in their abilities to perform eco-friendly tasks and innovate sustainably. Mastery experiences refer to repeated engagement in deliberate practice activities that lead to improved performance and increased belief in one's capabilities. As employees engage in EDP, they continuously refine their eco-innovation skills, resulting in higher levels of self-efficacy.

These mastery experiences are iterative and intentional, as individuals continuously assess their performance, seek feedback, and adjust their behavior to improve further. This process is supported by POS and DL, which provide the necessary resources, guidance, and encouragement for employees to persist in their efforts. Through this cycle, employees become more adept at eco-innovation tasks, strengthening their belief in their capacity to influence organizational sustainability. This belief, in turn, motivates them to engage in further deliberate practice, creating a positive feedback loop where *self-efficacy is reinforced and performance is enhanced*.

The mediating role of EDP in the proposed model operates through this mechanism of mastery experiences and self-efficacy. POS and DL function as external resources that create an environment conducive to the accumulation of these experiences, thus enabling employees to transform their efforts into eco-innovation outcomes.

Third, we propose the moderating role of ER between EDP and EP. Insights for this relationship are built on the social cognitive theory, which posits that high levels of self-efficacy may cause performance depreciation due to automization and routinization of the activity. That is to say, when individuals continue to put efforts in the existing tasks, they develop feelings of overconfidence, which is negatively linked with performance outcomes [19]. Hence, ER is an important resource that can allow cognitive modifications in oneself, thereby minimizing the likelihood of getting trapped in the arrested development. By examining the role of ER on the EDP–EP, our study not only benefits deliberate practice literature in the work context [17, 19, 39, 49] but also extends its implications in the general domains (*e.g.*, sports and music). We suggest that individuals possessing higher resilience will be more capable of stimulating high levels of self-efficacy for the smooth transition of deliberate practice into augmented performance.

## Practical implications

The findings of this study offer several practical insights for organizations aiming to improve eco-innovation performance (EP) through eco-friendly deliberate practice (EDP). Organizations must create supportive environments that encourage employees to engage in deliberate eco-innovation efforts. Both perceived organizational support (POS) and developmental leadership (DL) play crucial roles in promoting EDP and, in turn, EP [14, 16]. To facilitate these outcomes, companies can implement targeted interventions.

First, to raise employees' perceptions of organizational support, companies should implement strategies that promote *role clarity* [84]. Regular meetings where managers communicate the alignment between employees' eco-innovation efforts and the organization's sustainability goals can enhance role clarity. Additionally, fostering *job embeddedness*—the sense of belonging within the organization—can be achieved by establishing eco-friendly work groups or committees, allowing employees to collaborate on sustainability initiatives [85]. Moreover, offering *flexible working arrangements* [86], such as remote work or flexible hours, gives employees the opportunity to engage in eco-innovation activities, like exploring new technologies or improving eco-friendly processes. Structured programs such as "Green Fridays," where employees dedicate time to sustainability projects, can promote deliberate practice.

Second, leadership development is critical in supporting EDP. Organizations should invest in *leadership development programs* that equip managers to mentor and coach employees in eco-innovation skills [47]. Creating mentorship structures where leaders provide regular feedback and guidance on eco-innovation projects can be beneficial. For example, conducting *eco-performance reviews* where managers assess employees' sustainability contributions and offer personalized feedback can enhance employee engagement in eco-innovation. Furthermore, allowing employees more *job autonomy* [87] encourages ownership of their eco-innovation efforts, enabling them to experiment with new eco-friendly techniques, supported by leadership.

Third, given the moderating role of *employee resilience* (ER) in the EDP-EP relationship, organizations should consider implementing resilience-building programs. Resilience training can include workshops on stress management, adaptive thinking, and problem-solving, equipping employees to overcome challenges during eco-innovation projects. For instance, organizations could introduce a *Resilience Development Program* offering seminars and e-learning modules to help employees build strategies for bouncing back from setbacks [53]. These programs can prepare employees to maintain their eco-innovation efforts even when facing difficulties.

Lastly, organizations should emphasize *continuous learning* by integrating eco-innovation targets into performance reviews [39]. A company might introduce an award system recognizing employees who contribute innovative eco-friendly solutions, such as a "Green Innovator of the Month." Additionally, creating *innovation labs* or hosting sustainability hackathons can encourage employees to engage in green initiatives aligned with the organization's strategic goals.

By implementing these strategies—role clarity, leadership development, resilience training, and continuous learning—organizations can foster an environment conducive to EDP, ultimately enhancing their eco-innovation performance.

## Limitations and future directions

There are several limitations in the study. First, the authors examined the role of POS and DL on EP through the mediating role of EDP by employing a time-lagged, multisource design. Although all the survey items were not tapped at a single point, minimizing the issues of CMB. Future studies should utilize a longitudinal research design to investigate the hypothesized model. Second, only recently researchers have started examining the role of deliberate practice in employee-related outcomes. Therefore, the antecedents and outcomes of deliberate practice in the work context still need to be explored. Third, the study finds that EDP partially mediate the links between POS and EP and DL and EP, which indicates that the direct associations between POS and EP and DL and EP may also be influenced by some other contingent factors. Fourth, we invite future studies to outstretch the boundary conditions of the EDP–EP link by exploring other moderating variables. Last, the study was conducted in a non-Western cultural context, which warrants future studies to test these relationships in the Western context.

## Conclusion

In conclusion, this study highlights the critical role of POS and DL in driving EDP, which in turn enhances EP. The findings demonstrate that EDP acts as a key mediator between POS, DL, and EP, while ER strengthens the relationship between EDP and EP, particularly at higher levels of resilience. These insights underscore the importance of fostering supportive leadership, encouraging deliberate eco-friendly practices, and building resilience to promote sustainable innovation in organizations.

## Supporting information

**S1 Appendix. Measurement scales.**
(DOCX)

**S1 Data. 383 final data for analysis.**
(CSV)

## Author Contributions

**Conceptualization:** Yin-shi Jin, Asia Sohail, Shahid Iqbal, Tehreem Fatima, Arslan Ayub.

**Data curation:** Shahid Iqbal, Tehreem Fatima, Arslan Ayub.

**Formal analysis:** Asia Sohail.

**Funding acquisition:** Yin-shi Jin.

**Investigation:** Yin-shi Jin, Shahid Iqbal, Tehreem Fatima, Arslan Ayub.

**Methodology:** Yin-shi Jin, Shahid Iqbal, Tehreem Fatima, Arslan Ayub.

**Resources:** Asia Sohail.

**Visualization:** Asia Sohail.

**Writing – original draft:** Yin-shi Jin, Asia Sohail, Shahid Iqbal, Tehreem Fatima, Arslan Ayub.

**Writing – review & editing:** Asia Sohail.

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
