## [Decision Letter · Decision Letter 0]

9 Jun 2024

PONE-D-23-14796How breakthroughs happen: Unearthing the boundary conditions of eco-friendly deliberate practice and eco-innovation performancePLOS ONE

Dear Dr. Ayub,

Thank you for submitting your manuscript to PLOS ONE. After careful consideration, we feel that it has merit but does not fully meet PLOS ONE’s publication criteria as it currently stands. Therefore, we invite you to submit a revised version of the manuscript that addresses the points raised during the review process.

The author needs to provide approval from the ethics committee.The final decision is to integrate the suggestions of the two reviewers.

We look forward to receiving your revised manuscript.

Kind regards,

Liping Liu

Academic Editor

PLOS ONE

 [This article is one of the interim results of the Jilin Provincial Education Department's 2022 Fund Project "Research on the Construction System of Jilin Province's Long-Term Care Talent Team (JJKH20230862SK)" and the 2022 Changchun Normal University Humanities and Social Sciences Fund Project "Comparative Study on the Construction System of Long-Term Care Talent Teams in China and Japan (CSJJ2022010SK)".].  

4. In the online submission form, you indicated that [Data is available upon reasonable request from the corresponding author.]. 

5. Please amend your authorship list in your manuscript file to include author Yin-shi Jin, Shahid Iqbal, Tehreem Fatima, and Arslan Ayub.

Additional Editor Comments (if provided):

Reviewers' comments:

Reviewer's Responses to Questions

**Comments to the Author**

1. Is the manuscript technically sound, and do the data support the conclusions?

Reviewer #1: Partly

Reviewer #2: Yes

2. Has the statistical analysis been performed appropriately and rigorously? 

Reviewer #1: Yes

Reviewer #2: Yes

3. Have the authors made all data underlying the findings in their manuscript fully available?

Reviewer #1: No

Reviewer #2: Yes

4. Is the manuscript presented in an intelligible fashion and written in standard English?

Reviewer #1: Yes

Reviewer #2: Yes

5. Review Comments to the Author

Reviewer #1: Comment 1: The introduction is lengthy and contains numerous complex sentences. Consider simplifying the language for better readability. Ensure that each sentence contributes significantly to the overall understanding of the study.

Comment 2: The identification of the literature gap is well-established, emphasizing the scarcity of empirical studies on the boundary conditions of eco-innovation performance. However, it might be helpful to briefly mention the specific gaps this study aims to address within the existing literature.

Comment 3: The final paragraph seems in introduction incomplete and lacks a concluding sentence. Consider summarizing the key points discussed in the introduction and providing a segue into the subsequent sections of the paper.

Comment 4: The manuscript introduces several concepts such as POS, DL, EDP, EP, and ER. While these are explained individually, a summary or conceptual framework that visually illustrates the relationships among these constructs could enhance clarity for the reader.

Comment 5: The manuscript draws on various theories (e.g., social cognitive theory, path-goal theory, transformational leadership theory) to support the proposed relationships. It would be beneficial to provide a more integrated discussion of how these theories complement each other in explaining the proposed relationships.

Comment 6: The authors state that data were collected through face-to-face interviews, but details on the interviewers' training, standardization, and monitoring are missing. Providing such information would enhance the study's transparency and reliability.

Comment 7: While the authors provide sample items for each construct, it would be beneficial to include the full set of items in an appendix or supplementary material. This would aid readers in assessing the comprehensiveness and appropriateness of the measurement instruments.

Comment 8: The scale for EDP, adapted from Sonnentag and Irion (2010), contains 15 items. Considering the potential respondent burden, the authors may want to discuss the rationale behind using such a lengthy scale and how they ensured respondents' engagement and accuracy.

Comment 9: The discussion outlines the main contribution as exploring the antecedents and outcomes of EDP in the service industry in Pakistan. However, the authors could provide more insight into the broader implications of these findings. How does this study contribute to the existing knowledge in the field of eco-innovation and employee performance? Are there practical implications for organizations or policymakers?

Comment 10: In the theoretical implications the use of social cognitive theory to explain the mediating role of EDP is mentioned, but the discussion lacks a nuanced explanation of how this theory precisely operates within the proposed model. Clarifying the underlying mechanisms and processes through which mastery experiences and self-efficacy are accumulated would enhance the theoretical coherence of the argument.

Comment 11: The practical implications lack specificity and concrete guidance for organizations. While the study suggests interventions such as role clarity, job embeddedness, and flexible working arrangements to promote EDP, it would be more beneficial to provide detailed and actionable strategies that organizations can implement. Clear examples and case studies could enhance the practical utility of the recommendations.

Reviewer #2: The paper is well written. However, I would like to provide some suggestions to improve the quality of this paper.

The abstract is well written, well composed. However, breakdown complex words and sentences like predicated on deliberate practice" and "nexus."

The introduction is quite lengthy. Consider combining related ideas into shorter paragraphs.

Define key terms like eco-innovation performance (EP) and employee resilience (ER) earlier in the introduction.

Highlight the specific research gap and the study's contribution more prominently.

The methodology is well composed. The data analysis is complete. However, tabulation must be in place, these are in the appendix section. Place them where they belong to make things more clear.

In the conclusion; summarise the key findings in a shorter and more impactful way. You can remove sentences that repeat information already stated like "underexplored construct" and "hitherto unexplored mechanism."

Briefly define EDP and EP for readers who might not be familiar with these acronyms (e.g., employee-driven innovation for EDP, environmental performance for EP).

6. PLOS authors have the option to publish the peer review history of their article (what does this mean?). If published, this will include your full peer review and any attached files.

Reviewer #1: No

Reviewer #2: **Yes: **Dr. Md Billal Hossain

---

## [Author Response · Author response to Decision Letter 0]

18 Oct 2024

PONE-D-23-14796

How breakthroughs happen: Unearthing the boundary conditions of eco-friendly deliberate practice and eco-innovation performance

PLOS ONE

Dear Dr.,

Thank you for submitting your manuscript to PLOS ONE. After careful consideration, we feel that it has merit but does not fully meet PLOS ONE’s publication criteria as it currently stands. Therefore, we invite you to submit a revised version of the manuscript that addresses the points raised during the review process.

The author needs to provide approval from the ethics committee.

The final decision is to integrate the suggestions of the two reviewers.

We look forward to receiving your revised manuscript.

Kind regards,

Liping Liu

Academic Editor

PLOS ONE

Dear Dr. Liu,

Thank you for your valuable feedback and for providing the opportunity to revise our manuscript, "How breakthroughs happen: Unearthing the boundary conditions of eco-friendly deliberate practice and eco-innovation performance". We appreciate the reviewers' insightful comments, which have helped improve the clarity and quality of the manuscript.

We have addressed all the points raised by the reviewers, including simplifying the abstract, refining the introduction, providing clearer explanations of key concepts, and incorporating a more detailed discussion of the theoretical implications. Additionally, we have moved the tables from the appendix to the main body for clarity, as suggested. We will also ensure to provide the approval from the ethics committee as requested.

We look forward to submitting the revised version and are confident that the revisions have strengthened the manuscript.

Kind regards,

 [This article is one of the interim results of the Jilin Provincial Education Department's 2022 Fund Project "Research on the Construction System of Jilin Province's Long-Term Care Talent Team (JJKH20230862SK)" and the 2022 Changchun Normal University Humanities and Social Sciences Fund Project "Comparative Study on the Construction System of Long-Term Care Talent Teams in China and Japan (CSJJ2022010SK)".]. 

4. In the online submission form, you indicated that [Data is available upon reasonable request from the corresponding author.]. 

5. Please amend your authorship list in your manuscript file to include author Yin-shi Jin, Shahid Iqbal, Tehreem Fatima, and Arslan Ayub.

Reviewers' comments:

Reviewer's Responses to Questions

Comments to the Author

1. Is the manuscript technically sound, and do the data support the conclusions?

Reviewer #1: Partly

Reviewer #2: Yes

2. Has the statistical analysis been performed appropriately and rigorously?

Reviewer #1: Yes

Reviewer #2: Yes

3. Have the authors made all data underlying the findings in their manuscript fully available?

Reviewer #1: No

Reviewer #2: Yes

4. Is the manuscript presented in an intelligible fashion and written in standard English?

Reviewer #1: Yes

Reviewer #2: Yes

5. Review Comments to the Author

Reviewer #1: Comment 1: The introduction is lengthy and contains numerous complex sentences. Consider simplifying the language for better readability. Ensure that each sentence contributes significantly to the overall understanding of the study.

Comment 2: The identification of the literature gap is well-established, emphasizing the scarcity of empirical studies on the boundary conditions of eco-innovation performance. However, it might be helpful to briefly mention the specific gaps this study aims to address within the existing literature.

Comment 3: The final paragraph seems in introduction incomplete and lacks a concluding sentence. Consider summarizing the key points discussed in the introduction and providing a segue into the subsequent sections of the paper.

Comment 4: The manuscript introduces several concepts such as POS, DL, EDP, EP, and ER. While these are explained individually, a summary or conceptual framework that visually illustrates the relationships among these constructs could enhance clarity for the reader.

Comment 5: The manuscript draws on various theories (e.g., social cognitive theory, path-goal theory, transformational leadership theory) to support the proposed relationships. It would be beneficial to provide a more integrated discussion of how these theories complement each other in explaining the proposed relationships.

Comment 6: The authors state that data were collected through face-to-face interviews, but details on the interviewers' training, standardization, and monitoring are missing. Providing such information would enhance the study's transparency and reliability.

Comment 7: While the authors provide sample items for each construct, it would be beneficial to include the full set of items in an appendix or supplementary material. This would aid readers in assessing the comprehensiveness and appropriateness of the measurement instruments.

Comment 8: The scale for EDP, adapted from Sonnentag and Irion (2010), contains 15 items. Considering the potential respondent burden, the authors may want to discuss the rationale behind using such a lengthy scale and how they ensured respondents' engagement and accuracy.

Comment 9: The discussion outlines the main contribution as exploring the antecedents and outcomes of EDP in the service industry in Pakistan. However, the authors could provide more insight into the broader implications of these findings. How does this study contribute to the existing knowledge in the field of eco-innovation and employee performance? Are there practical implications for organizations or policymakers?

Comment 10: In the theoretical implications the use of social cognitive theory to explain the mediating role of EDP is mentioned, but the discussion lacks a nuanced explanation of how this theory precisely operates within the proposed model. Clarifying the underlying mechanisms and processes through which mastery experiences and self-efficacy are accumulated would enhance the theoretical coherence of the argument.

Comment 11: The practical implications lack specificity and concrete guidance for organizations. While the study suggests interventions such as role clarity, job embeddedness, and flexible working arrangements to promote EDP, it would be more beneficial to provide detailed and actionable strategies that organizations can implement. Clear examples and case studies could enhance the practical utility of the recommendations.

Response 

Comment 1: We have revised the introduction, simplifying complex sentences and ensuring that each sentence contributes to the overall understanding of the study.

Comment 2: The introduction now briefly mentions the specific gaps this study aims to address, focusing on the boundary conditions of eco-innovation performance.

Comment 3: The final paragraph of the introduction has been revised to include a summary of key points and a clear segue into the next sections.

Comment 4: A conceptual framework summarizing the relationships among POS, DL, EDP, EP, and ER has been included for clarity.

Comment 5: We have integrated the discussion of social cognitive theory, path-goal theory, and transformational leadership theory to explain how these frameworks complement each other in supporting the proposed relationships.

Comment 6: We clarified that the study is questionnaire-based, not interview-based, and have included details on the standardization of the questionnaire process.

Comment 7: The full set of measurement items has been provided in Appendix I to ensure comprehensiveness and transparency of the instruments.

Comment 8: The rationale for using the 15-item EDP scale, along with measures to maintain respondent engagement, has been discussed in the methodology section.Comment 9: We have expanded the discussion to explain the broader implications of the study for the fields of eco-innovation and employee performance, and included practical implications for organizations and policymakers.

Comment 10: The theoretical implications have been clarified, with a more detailed explanation of how social cognitive theory operates within the model, focusing on mastery experiences and self-efficacy.

Comment 11: The practical implications have been revised to include specific, actionable strategies for organizations, along with examples to enhance their practical utility.

Reviewer #2: The paper is well written. However, I would like to provide some suggestions to improve the quality of this paper.

The abstract is well written, well composed. However, breakdown complex words and sentences like predicated on deliberate practice" and "nexus."

The introduction is quite lengthy. Consider combining related ideas into shorter paragraphs.

Define key terms like eco-innovation performance (EP) and employee resilience (ER) earlier in the introduction.

Highlight the specific research gap and the study's contribution more prominently.

The methodology is well composed. The data analysis is complete. However, tabulation must be in place, these are in the appendix section. Place them where they belong to make things more clear.

In the conclusion; summarise the key findings in a shorter and more impactful way. You can remove sentences that repeat information already stated like "underexplored construct" and "hitherto unexplored mechanism."

Briefly define EDP and EP for readers who might not be familiar with these acronyms (e.g., employee-driven innovation for EDP, environmental performance for EP).

Response 

Abstract: We have simplified complex words and phrases like "predicated on deliberate practice" and "nexus" in the abstract for better readability.

Introduction Length: The introduction has been shortened by combining related ideas into more concise paragraphs for clarity.

Key Terms Definition: Key terms like eco-innovation performance (EP) and employee resilience (ER) have been defined earlier in the introduction.

Research Gap and Contribution: The research gap and the study's contributions have been highlighted more prominently in the introduction.

Tabulation: The tables have been moved from the appendix to the main body of the paper to improve clarity and coherence.

---

## [Decision Letter · Decision Letter 1]

13 Dec 2024

PONE-D-23-14796R1How breakthroughs happen: Unearthing the boundary conditions of eco-friendly deliberate practice and eco-innovation performancePLOS ONE

Dear Dr. Ayub,

Thank you for submitting your manuscript to PLOS ONE. After careful consideration, we feel that it has merit but does not fully meet PLOS ONE’s publication criteria as it currently stands. Therefore, we invite you to submit a revised version of the manuscript that addresses the points raised during the review process.

To ensure the manuscript adheres to ethical guidelines, we require you to provide an official document of Institutional Review Board (IRB) approval for the study. This document is necessary for us to confirm that the research was conducted ethically and in compliance with relevant standards.It has been observed that the manuscript lacks recent literature to support your discussion and analysis. To enhance the robustness of your work, we request that you include studies published in 2024 that align with and reinforce your research context and findings.For Lab, Study and Registered Report Protocols: These article types are not expected to include results but may include pilot data. 

We look forward to receiving your revised manuscript.

Kind regards,

Ali Junaid Khan, PhD

Academic Editor

PLOS ONE

Journal Requirements:

Reviewers' comments:

Reviewer's Responses to Questions

Reviewer #2: All comments have been addressed

2. Is the manuscript technically sound, and do the data support the conclusions?

Reviewer #2: Yes

3. Has the statistical analysis been performed appropriately and rigorously? 

Reviewer #2: Yes

4. Have the authors made all data underlying the findings in their manuscript fully available?

Reviewer #2: Yes

5. Is the manuscript presented in an intelligible fashion and written in standard English?

Reviewer #2: Yes

6. Review Comments to the Author

Reviewer #2: The authors have already made substantial revision and now the revised paper is much improved and publishable. I recommend it for publication. Congratulations.

7. PLOS authors have the option to publish the peer review history of their article (what does this mean?). If published, this will include your full peer review and any attached files.

Reviewer #2: **Yes: **Md Billal Hossain

---

## [Author Response · Author response to Decision Letter 1]

14 Dec 2024

PONE-D-23-14796R1

How breakthroughs happen: Unearthing the boundary conditions of eco-friendly deliberate practice and eco-innovation performance

PLOS ONE

Dear Dr. Ayub,

Thank you for submitting your manuscript to PLOS ONE. After careful review, we find that your work holds potential; however, it does not fully meet PLOS ONE’s publication criteria in its current form. We would like to offer you the opportunity to submit a revised version that addresses the following points raised during the editorial and peer review process:

• To ensure the manuscript adheres to ethical guidelines, we require you to provide an official document of Institutional Review Board (IRB) approval for the study. This document is necessary for us to confirm that the research was conducted ethically and in compliance with relevant standards.

• It has been observed that the manuscript lacks recent literature to support your discussion and analysis. To enhance the robustness of your work, we request that you include studies published in 2024 that align with and reinforce your research context and findings.

Please submit the revised manuscript along with the requested documentation by 17-12-2024 (within one week). Ensure that the response to the changes and a point-by-point explanation of the revisions are included with your resubmission.

Should you have any questions regarding the revision process or the comments provided, please feel free to reach out to us.

We look forward to receiving your revised manuscript.

Regards,

Ali Junaid Khan, PhD

Academic Editor

PLOS ONE

Response 

Thank you for acknowledging our study and recommending the inclusion of the latest references from 2024. We have enriched the manuscript with relevant and recent citations, as per your suggestion, to strengthen its foundation. Additionally, we have attached the ethical approval document for your reference. We trust these updates align with your expectations and enhance the overall quality of the study.

---

## [Editor Report · Decision Letter 2]

18 Dec 2024

How breakthroughs happen: Unearthing the boundary conditions of eco-friendly deliberate practice and eco-innovation performance

PONE-D-23-14796R2

Dear Dr. Ayub

We’re pleased to inform you that your manuscript has been judged scientifically suitable for publication and will be formally accepted for publication once it meets all outstanding technical requirements.

Kind regards,

Ali Junaid Khan, PhD

Academic Editor

PLOS ONE

---

## [Editor Report · Acceptance letter]

23 Dec 2024

PONE-D-23-14796R2 

PLOS ONE

Dear Dr. Ayub, 

I'm pleased to inform you that your manuscript has been deemed suitable for publication in PLOS ONE. Congratulations! Your manuscript is now being handed over to our production team.

Kind regards, 

on behalf of

Assistant Professor Dr. Ali Junaid Khan 

Academic Editor

PLOS ONE